META-RESEARCH

# A 10-year follow-up study of sex inclusion in the biological sciences

**Abstract** In 2016, to address the historical overrepresentation of male subjects in biomedical research, the US National Institutes of Health implemented a policy requiring investigators to consider sex as a biological variable. In order to assess the impact of this policy, we conducted a bibliometric analysis across nine biological disciplines for papers published in 34 journals in 2019, and compared our results with those of a similar study carried out by Beery and Zucker in 2009. There was a significant increase in the proportion of studies that included both sexes across all nine disciplines, but in eight of the disciplines there was no change in the proportion studies that included data analyzed by sex. The majority of studies failed to provide rationale for single-sex studies or the lack of sex-based analyses, and those that did relied on misconceptions surrounding the hormonal variability of females. Together, these data demonstrate that while sex-inclusive research practices are more commonplace, there are still gaps in analyses and reporting of data by sex in many biological disciplines.

**NICOLE C WOITOWICH\*, ANNALIESE BEERY AND TERESA WOODRUFF**

## Introduction

Studies of both males and females are essential to the advancement of human health, and the influences of sex on the prevalence, presentation, and progression of many disease states is profound. Yet, within the biological sciences, it has been a common and preferential practice to utilize male research subjects in basic and pre-clinical research (*Beery and Zucker, 2011*; *Kong et al., 2016*; *Sugimoto et al., 2019*; *Yoon et al., 2014*). This male bias stems from the misconception that female animals increase experimental variability due to cyclical fluctuating hormones and the historical belief that no major differences exist between the sexes outside of reproductive functions (*Institute of Medicine, 2001*). These biases are not limited to the basic sciences, but extend into clinical research as well (*Geller et al., 2018*; *Mansukhani et al., 2016*; *Prakash et al., 2018*; *Scott et al., 2018*).

Initial reports calling for the inclusion of females in research and which describe the limitations of sex-biased studies began in the 1990s and extended in to the early 2000s (*Berkley, 1992*; *Holdcroft, 2007*; *Mogil and Chanda, 2005*). In 2009, Beery and Zucker conducted a multi-disciplinary review of primary literature which quantified the extent of sex-bias across several research areas in the biological sciences (*Beery and Zucker, 2011*). Since that report, there have been numerous calls to address this issue through sex-inclusive research practices and policies (*Kim et al., 2010*; *Klein et al., 2015*; *Mazure and Jones, 2015*; *Woodruff, 2014*), culminating in 2016 when the National Institutes of Health (NIH) in the United States implemented a policy requiring investigators to consider sex as a biological variable (*Clayton and Collins, 2014*). The intent of the policy is to ensure equal representation of males and females in vertebrate research studies, unless there is significant justification to support the use of a single-sex. Many lauded the policy (*Mogil, 2016*; *Shansky and Woolley, 2016*), yet there were still those who saw it as unnecessary and feared that it would be time consuming,

\*For correspondence: nicole. woitowich@northwestern.edu

**Competing interests:** The authors declare that no competing interests exist.

costly, increase experimental variability, and require expertise in the study of sex differences (*Woitowich and Woodruff, 2019*). Considering sex as a biological variable does not require investigators to power studies in order to determine sex differences nor does it ask investigators to analyze data by sex. Yet, these common misconceptions persist, despite clarifications and guidance surrounding sex-inclusive research practices (*Arnegard et al., 2020*; *Becker et al., 2016*; *Clayton, 2018*; *Miller et al., 2017*; *Shansky, 2019*). Recently, several studies have monitored the progress of sex-inclusive research practices following NIH policy implementation in the fields of microbiology and immunology (*Potluri et al., 2017*), as well as neuroscience (*Mamlouk et al., 2020*) utilizing methodologies similar to Beery and Zucker. Here, we present a 10 year follow-up study to the initial Beery and Zucker report by conducting a systematic review to assess sex-inclusive research practices within nine of the biological disciplines and 34 of the scholarly journals originally surveyed in 2009. We provide an updated perspective on the state of sex-inclusive research within the biological sciences, and highlight areas of improvement alongside shortcomings in the decade since Beery and Zucker conducted their original study.

## Results

In 2009, Beery and Zucker conducted a bibliometric analysis of 841 articles from high-impact journals, across ten biological disciplines which quantified the extent of male-bias in research and a noted lack of sex-based analyses when males and females were both included as research subjects (*Beery and Zucker, 2011*). We recapitulated the work of Beery and Zucker utilizing a similar bibliometric analysis of 720 journal articles, corresponding to nine of the original disciplines and 34 journals surveyed in 2009 (*Table 1*).

### Subject sex across disciplines

In 2019, 49% (n = 356) of studies reported using both male and female research subjects, resulting in a significant increase in sex inclusion demographics compared to 28% of articles surveyed 2009 (n = 232, p<0.0001; *Figure 1A*). Six of the nine disciplines demonstrated a significant increase in the use of both sexes (*Figure 1*). Between 2009 and 2019, the largest increases in sex-inclusive studies were seen in the fields of neuroscience (29% vs. 63%, p<0.0001) and immunology (16% vs. 46%, p<0.0001), followed by endocrinology (30% vs. 56%, p=0.001), general biology (34% vs. 59%, p=0.002), physiology (13% vs. 36%, p=0.001), and behavioral physiology (43% vs. 61%, p=0.018). In reproduction,

**Table 1.** Journals surveyed by subject area in 2009 and 2019.

| Discipline | Journal A | Journal B | Journal C | Journal D |
|---|---|---|---|---|
| **General Biology** | PLoS Biology | Proceedings of the Royal Society B: Biological Sciences | Nature | Science |
| **Immunology** | Journal of Immunology | Infection and Immunity | Immunity | Vaccine |
| **Neuroscience** | Journal of Neuroscience | Neuroscience | The Journal of Comparative Neurology | Nature Neuroscience |
| **Physiology** | Journal of Physiology (London) | American Journal of Physiology – Renal Physiology | American Journal of Physiology – Gastrointestinal and Liver Physiology | American Journal of Physiology – Heart and Circulatory Physiology |
| **Pharmacology** | Neuropsycho-pharmacology | Journal of Psychopharma-cology | The Journal of Pharmacology and Experimental Therapeutics | British Journal of Pharmacology |
| **Reproduction** | Biology of Reproduction | Reproduction | | |
| **Endocrinology** | European Journal of Endocrinology | Journal of Neuroendo-crinology | Endocrinology | American Journal of Physiology – Endocrinology and Metabolism |
| **Behavioral Physiology** | Journal of Comparative Psychology | Behavioral Neuroscience | Physiology and Behavior | Hormones and Behavior |
| **Behavior** | Behavioral Ecology and Sociobiology | Animal Behaviour | Animal Cognition | Behavioral Ecology |

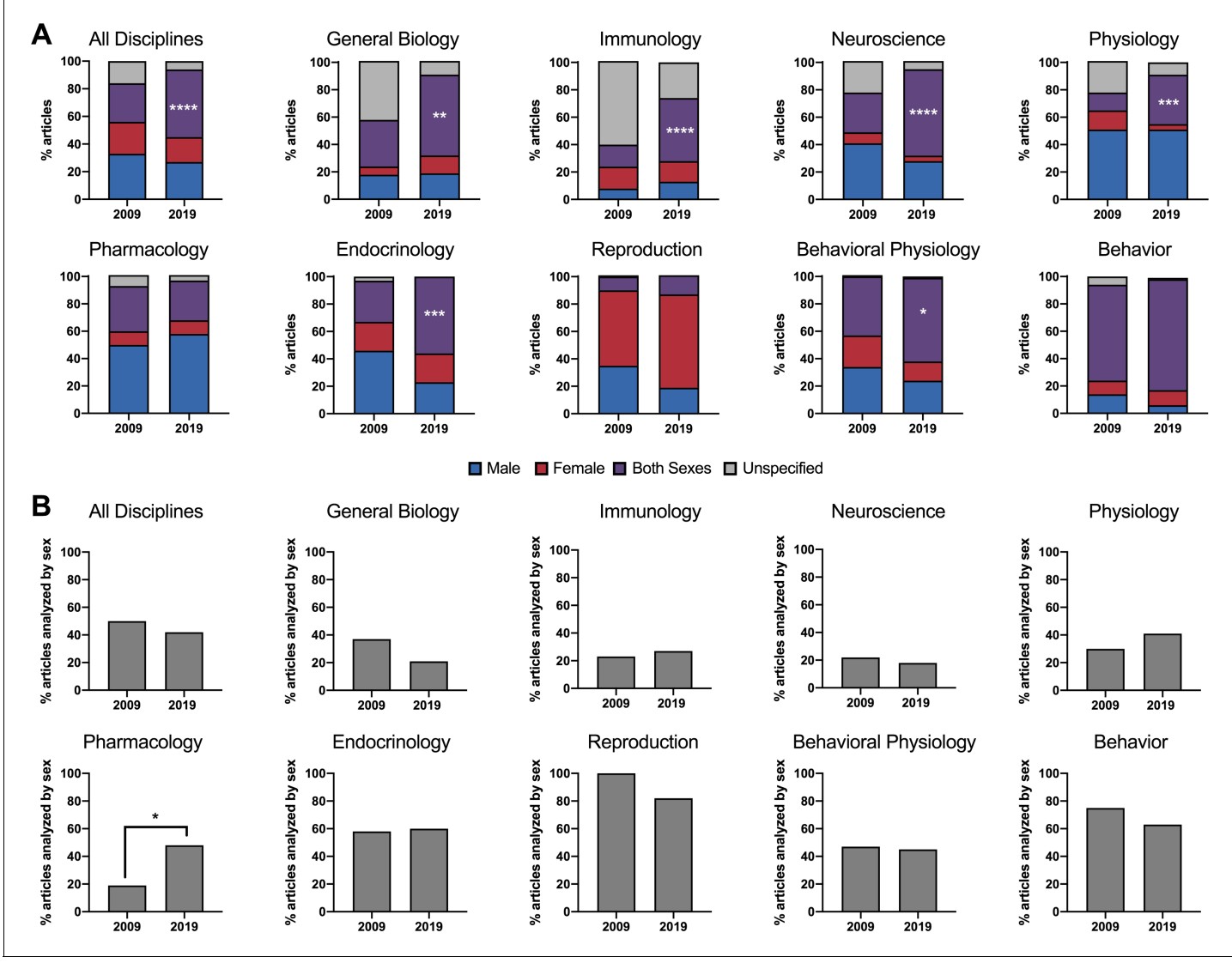

**Figure 1.** Comparison of studies by field, sex, and sex-based analyses in 2009 and 2019. (A). The proportion of articles surveyed in 2009 and 2019 which utilized male subjects, female subjects, both male and female subjects, or those that did not specify the sex of the subjects. Data are presented by individual biological discipline as well as by the sum of all nine disciplines. (B). The percentage of articles surveyed in 2009 and 2019 which utilized both male and female subjects and conducted sex-based analyses, either by including sex as a covariate or by subgroup analyses. Data are presented by individual biological discipline as well as by the sum of all nine disciplines. The source data for this figure are in *Supplementary file 1*.

single-sex studies remained the norm, and studies of both males and females increased only marginally (10% vs. 14%, p=0.35), while the number of female only research studies increased, corresponding to a female to male subject ratio of 1.6:1 in 2009 to 3.6:1 in 2019. Behavior remained the most inclusive biological discipline with 70% and 81% of studies reporting the use of both sexes in 2009 and 2019, respectively, largely driven by sex-inclusive field studies. Pharmacology was the only field to trend downward with 29% of articles reporting the use of both sexes in 2019 compared to 33% in 2009

(p=0.607). Likewise, there was an increase in the male to female subject ratio from 5:1 in 2009 to 5.8:1 in 2019.

### Sex based analyses by discipline
For articles that reported the inclusion of both sexes in 2019, data were collected on whether or not the authors conducted sex-based analyses. Out of 356 of the journal articles which used both sexes in 2019, only 42% analyzed data by sex, compared to 50% in 2009 (n = 117, p=0.3; *Figure 1B*). Pharmacology was the only biological discipline to demonstrate a significant

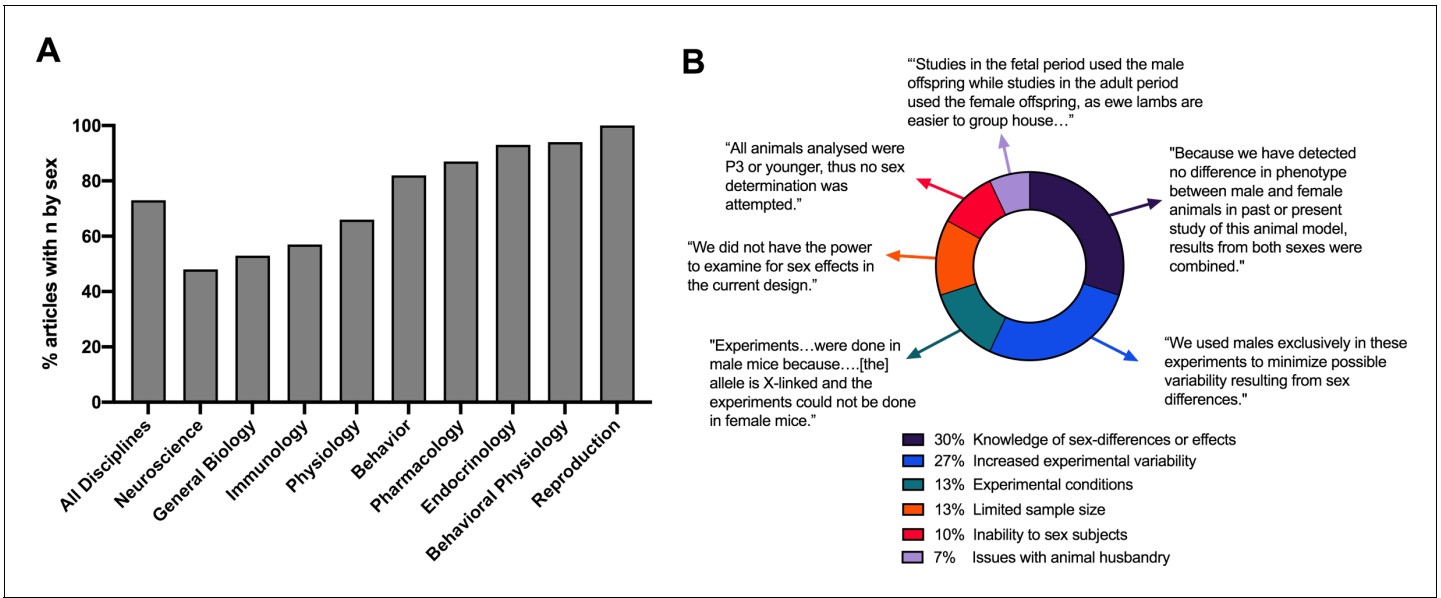

**Figure 2.** Percent of articles which provided the sample size (n) by sex, rationale for single-sex studies, or rationale for the lack of sex-based analyses. (A). The percentage of articles which utilized both male and female subjects and provided a description of the sample size by sex. Data are presented by individual biological discipline as well as by the sum of all nine disciplines. (B). Categorization of articles which provided rationale for single sex-studies or the lack of sex-based analyses (n = 30) into seven distinct themes. Each theme includes representative rationale derived from the experimental methods. The source data for this figure are in *Supplementary file 1*.

increase in sex-based analyses from 19% in 2009 to 48% in 2019 (p=0.033; *Figure 1B*).

### Description of sample size by sex across disciplines

For articles that reported the inclusion of both sexes in 2019, data were collected on whether the authors provided a description of the sample size (n) by sex. Out of the 356 articles that used both sexes, 27% failed to provide a description of the sample size by sex (*Figure 2A*). Neuroscience articles failed to provide a description of the sample size by sex 52% (n = 26) of the time, along with general biology at 47% (n = 22) and immunology at 43% (n = 19).

### Rationale for single sex studies or lack of sex-based analyses

For all 720 articles analyzed in 2019, data were collected on whether the authors provided a justification for the use of a single sex or rationale for the lack of sex-based analyses. Thirty articles included a range of explanations related to sex-inclusion and sex-based analyses (*Figure 2B*). Justifications for single sex studies included: a priori knowledge of sex-differences or sex-specific effects (n = 9), the potential for increased experimental variability (n = 8), experimental conditions which limited the use of both sexes (n = 4) and difficulties in animal husbandry

(n = 2). Rationale for the lack of sex-based analyses included: limited sample sizes to determine statistical significance (n = 4) or an inability to determine the sex of the subject (n = 3). Only two studies specified that the authors did not identify any sex differences, so the dataset was analyzed in aggregate.

## Discussion

Notably, the number of sex-inclusive research studies has significantly increased across most biological disciplines. At face value, this change is encouraging and suggests that the scientific community may have an increased awareness and understanding of the need for sex-inclusive research and its contribution to experimental rigor and reproducibility (*Clayton, 2018*; *Miller et al., 2017*). At the same time, close to one third of all research studies that utilized both male and female subjects failed to quantify their sample size by sex. Ironically, this is most prevalent in the fields which reported the greatest increases in sex-inclusive research (ex. neuroscience, immunology, and general biology) At best, this result indicates that investigators may not think it is important to provide a description of the sample size by sex in the absence of sex-based analyses. In a less ideal case, the representation of males and females is not well

balanced, and this may be intentionally obscured. Single-sex studies are valid and warranted, provided there is evidence-based rationale for the case. Yet, several studies explicitly stated that they excluded both sexes as a means to prevent experimental variability, which is an erroneous belief and unsound research practice (*Beery, 2018*; *Prendergast et al., 2014*).

Perhaps most concerning, improvements in the inclusion of both sexes over the past decade have not been accompanied by general improvement in sex-based analyses, despite repeated calls and guidelines for such analyses (*Beltz et al., 2019*; *Clayton, 2018*; *Clayton and Collins, 2014*; *Hankivsky et al., 2018*; *Prager, 2017*). Sex-based analyses may uncover sex differences for a given trait, prompting the development of sex-specific prevention strategies, drug targets, or other therapies beneficial to both sexes (*Yang et al., 2019*). And while it is reasonable to aggregate and analyze data from both sexes if it has been established that there are no sex-differences for a given trait or condition, out of the 720 articles reviewed here, only two conveyed this information in their methods. When this information is lacking, the reader is tasked with making the assumption that either there are no sex differences or that sex-differences have yet to be examined. In either case, this can lead to redundant research efforts requiring additional time, money, and biological resources.

The data presented here highlight a continued need for education, awareness, and advocacy surrounding sex-based research practices including the consideration of sex as a biological variable. We call upon academic publishers to require a description of sex, rationale for single-sex studies or lack of sex-based analyses in the experimental methods. In the absence of formal policies, reviewers can ask for these essential criteria. In addition, funders can also contribute to the advancement of rigorous sex-inclusive science by requiring grant proposals to include appropriate sex-based reporting and analyses and determine funding success on the evaluation of sex and other key biological variables. Lastly, we call upon universities to encourage the consideration of sex as a biological variable through institutional review boards (IRBs) and institutional animal care and use committees (IACUC) oversight (*Duffy et al., 2020*) and by providing instruction to biomedical trainees on sex-inclusion, reporting, and analyses through established responsible conduct of research modules and within medical school curricula. Only

together, through concerted, tripartite efforts at the institutional, funder, and publishing levels will the consideration of sex as a biological variable become standard practice (*Tannenbaum et al., 2019*). Together, this will allow us to improve our understanding of health and disease for both men and women and to further the reality of personalized medicine.

## Methods

A systematic sampling of journal articles from 2019 was conducted using the methodologies originally described in *Beery and Zucker, 2011*. All articles were reviewed and coded by one of us (NCW) in order to minimize coding bias. Briefly, journal articles were assessed for sex-inclusive research practices from nine biological disciplines and 34 journals sampled by *Beery and Zucker, 2011*. These disciplines included: General Biology, Immunology, Neuroscience, Physiology, Pharmacology, Reproduction, Endocrinology, Behavioral Physiology, and Behavior. Zoology, which was studied by Beery and Zucker, was excluded here due to a limited number of mammalian studies available to survey at the time of manuscript preparation. Four journals were selected to represent each discipline, with the exception of Reproduction (*Table 1*). For each journal, the first 20 primary research articles which met eligibility criteria were surveyed in 2019. For the two reproductive biology journals, the first 40 journal articles were surveyed for 2019. For the majority of disciplines, the first 20 research articles which met eligibility criteria were published between January and April of 2019, whereas articles from other disciplines were published between January through June (Endocrinology), August (Behavioral Physiology) and October (Behavior) of 2019.

The eligibility criteria for studies in this analysis were as follows.

Inclusion criteria (all criteria required): i) Reported use of any vertebrate mammal in some part of the experimental methods, including those which describe the generation of primary cell culture; ii) Published after January 1 st, 2019; iii) Published in the English language.

Exclusion criteria (each criterion can exclude): i) Type of article: review articles, brief communications, or viewpoints; ii) Articles published in a special or themed issue; iii) Reports utilizing fetal organisms or those restricted to immortal cell lines.

When journals were arranged by subtopics, articles were sampled evenly across several topics. In journals such as *Nature* and *Science*, only articles pertaining to the biological sciences were considered.

Articles were coded for sex. Sex was recorded as male, female, both sexes, or unspecified. Following the strategy of *Beery and Zucker, 2011*, coding was biased in favor of inclusivity and articles were categorized as using both sexes when different parts of a study utilized different sexes. Likewise, field studies were categorized as investigating both sexes when this was explicitly noted or could be inferred by the methods provided. Articles which utilized both sexes were further evaluated for a description of the sample size by sex and whether data were analyzed by sex, including sex as a covariate or subgroup analyses by sex. For all articles reviewed, we noted if the authors provided rationale for the use of a single sex or the lack of sex-based analyses.

Data analyses were primarily qualitative, with a small quantitative component. Descriptive statistics were used where appropriate. Nominal data were described as n (%). We compared the 2019 data to 2009 data in *Beery and Zucker, 2011*. Chi-squared tests were used to assess differences between the use of both sexes in 2009 compared to 2019, and the number of studies which analyzed data by sex in 2009 compared to 2019 (GraphPad Prism, version 7.0). p-values<0.05 were considered significant.

## Acknowledgements
We thank Dr Irving Zucker for his encouragement to conduct this study.

**Nicole C Woitowich** is in the Women's Health Research Institute and the Department of Obstetrics and Gynecology, Feinberg School of Medicine, Northwestern University, Chicago, United States

nicole.woitowich@northwestern.edu

https://orcid.org/0000-0002-3449-2547

**Annaliese K Beery** is in the Department of Psychology, the Department of Biology, and the Program in Neuroscience, Smith College, Northampton, United States

https://orcid.org/0000-0002-1249-9182

**Teresa K Woodruff** is in the Women's Health Research Institute and the Department of Obstetrics and Gynecology, Feinberg School of Medicine, Northwestern University, Chicago, United States

https://orcid.org/0000-0002-1197-3399

*Author contributions:* Nicole C Woitowich, Conceptualization, Data curation, Formal analysis, Supervision, Validation, Investigation, Visualization, Methodology, Writing - original draft, Project administration, Writing - review and editing; Annaliese Beery, Conceptualization, Data curation, Methodology, Writing - review and editing; Teresa Woodruff, Conceptualization, Methodology, Writing - review and editing

*Competing interests:* The authors declare that no competing interests exist.

### Funding

| Funder | Grant reference number | Author |
| --- | --- | --- |
| Northwestern University | Women's Health Research Institute | Nicole C Woitowich Teresa Woodruff |

The funders had no role in study design, data collection and interpretation, or the decision to submit the work for publication.

#### Decision letter and Author response
Decision letter https://doi.org/10.7554/eLife.56344.sa1
Author response https://doi.org/10.7554/eLife.56344.sa2

## Additional files

### Supplementary files
• Supplementary file 1. Data for *Figures 1* and *2*. This file contains the data represented in *Figures 1* and *2*, as well as p-values for all the statistical tests shown in *Figures 1* and *2*.

• Transparent reporting form

### Data availability
All data generated or analysed during this study are included in the manuscript and supporting files. Source data files have been provided for Figures 1 and 2.

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
