## [Decision Letter]

Your article has been reviewed by three peer reviewers, and the evaluation has been overseen by a Reviewing Editor (Cassidy Sugimoto) and the *eLife* Features Editor (Peter Rodgers). The following individuals involved in review of your submission have agreed to reveal their identity: Rebecca Shansky (Reviewer #2); Londa Schiebinger (Reviewer #3).

The reviewers and editors have discussed the reviews and we have drafted this decision to help you prepare a revised submission. We hope you will be able to submit the revised version within two months.

Summary:

This manuscript is a 10-year check in on a seminal paper (Beery & Zucker 2011) that systematically examined sex bias in biomedical animal research. Given the introduction of NIH's "Considering Sex as a Biological Variable" (SABV) mandate in 2016, it is important to assess whether research practices have changed over the past decade.

The authors conducted a survey of 2019 publications in ten major fields, and report that in some fields, inclusion of both sexes has increased but that for the most part this has not been accompanied by an increase in analysis of data by sex or reporting the n's of each sex used. The authors call for journals to require more detailed information about the sex of animals and justification of single sex studies. I especially appreciate the assessment of "excuses" for the single sex studies.

This study is a critical addition to the literature and will help biomedical scientists understand how well SABV is working and what needs to be done to make it even more successful.

This is an important article and it should be published once the following points have been addressed.

Essential revisions:

1) Introduction and Discussion: It was disappointing to see that the introduction and discussion did not address/include several key historical and recently published studies. These studies include some of the first studies that attempted to raise awareness regarding the lack of sex reporting (Berkley 1992; Mogil and Chanda 2005; both published before the initial 2009 study), updates to the 2009 study that robustly analyzed key fields that were documented as problematic in the 2009 study (Potluri et al., 2017; Mamlouk et al., 2020), and recently published arguments and analyses that are advancing sex-inclusive research (Shanskey, 2019; Galea et al., 2020). There are others as well (eg, Becker et al., 2016). While I am well aware that not every relevant paper can be cited/discussed, I encourage the authors to enhance the robustness of their literature search to help place their important results within the broader context.

2) I would encourage the authors to point out that SABV explicitly states that experiments need not be powered to detect sex differences or that data be analyzed by sex. However, their calls to require reporting of sex n's are well founded and will improve rigor and reproducibility.

3) The authors call upon academic publishers to require sex reporting. They should also call upon funders to require appropriate sex reporting and analysis in grant proposals--and make funding decision on the quality of that (and other) analysis. The authors should also call upon universities (medical schools) to teach sex reporting and analysis in the curriculum. In Tannenbaum et al (Nature 575: 137-146) the present reviewer [LS] and colleagues discuss the three pillars of the science infrastructure: Funding agencies, Universities, and Peer-reviewed journals. I encourage the authors here to add funders and universities to the paragraph where they discuss academic publishers.

4) Methods: "First 20 primary research articles" - Since only the first 20 articles were analyzed does this mean that most of the data was collected in studies published between January-March 2019? It would be a good idea to document this in the methods in case the data from 2019 need to be compared to future analyses that select across the entire year.

5) Methods: Interrater reliability: Did more than one person analyze studies? Were intra and inter-rater reliability controls performed to ensure accurate analysis?

6) Methods: "Organismal mammalian work" - This label is vague and could be divergently interpreted by different readers. Does this just mean whole organism behavioral analysis? Does it include all mammals, including primates and humans, or just rodents? Beyond answering these specific questions, the entire a priori article section criteria needs to be much better documented. This documentation should include how the employed protocol may potentially skew findings.

7) Methods: "Whether data were analyzed be sex." This definition is vague and requires further documentation. What analyses were considered an analysis by sex? For example, if sex was employed as a covariate, was this considered a sex analysis? What if the data or analysis/statistics were not shown? What if data were presented as disaggregated by sex but not further analyzed? Similar to the previous point, this section requires more robust documentation so that the reader can understand how this study was conducted.

---

## [Author Response]

[We repeat the reviewers’ points here in italic, and include our replies in Roman.]

Essential revisions:1) Introduction and Discussion: It was disappointing to see that the introduction and discussion did not address/include several key historical and recently published studies. These studies include some of the first studies that attempted to raise awareness regarding the lack of sex reporting (Berkley 1992; Mogil and Chanda 2005; both published before the initial 2009 study), updates to the 2009 study that robustly analyzed key fields that were documented as problematic in the 2009 study (Potluri et al., 2017; Mamlouk et al., 2020), and recently published arguments and analyses that are advancing sex-inclusive research (Shanskey, 2019; Galea et al., 2020). There are others as well (eg, Becker et al., 2016). While I am well aware that not every relevant paper can be cited/discussed, I encourage the authors to enhance the robustness of their literature search to help place their important results within the broader context.

We thank the reviewers for these suggestions and have enhanced our introduction to include a robust review of the literature related to the SABV policy and sex- and gender-inclusive research practices. Which, in turn, highlights the importance of this work by including additional historical context.

2) I would encourage the authors to point out that SABV explicitly states that experiments need not be powered to detect sex differences or that data be analyzed by sex. However, their calls to require reporting of sex n's are well founded and will improve rigor and reproducibility.

We have modified our introduction to ensure the clarification of this common misconception is included and provide more detailed information about the SABV policy.

3) The authors call upon academic publishers to require sex reporting. They should also call upon funders to require appropriate sex reporting and analysis in grant proposals--and make funding decision on the quality of that (and other) analysis. The authors should also call upon universities (medical schools) to teach sex reporting and analysis in the curriculum. In Tannenbaum et al (Nature 575: 137-146) the present reviewer [LS] and colleagues discuss the three pillars of the science infrastructure: Funding agencies, Universities, and Peer-reviewed journals. I encourage the authors here to add funders and universities to the paragraph where they discuss academic publishers.

We have added these additional “calls to action” and included recent work encouraging IRB and IACUC oversight as well (Duffy et al 2020).

4) Methods: "First 20 primary research articles" - Since only the first 20 articles were analyzed does this mean that most of the data was collected in studies published between January-March 2019? It would be a good idea to document this in the methods in case the data from 2019 need to be compared to future analyses that select across the entire year.

We thank the reviewers for this comment and have added this information into the Methods section. For the majority of fields, the articles analyzed here were published between January - April 2019, but some fields extended further into the calendar year, which we noted in the Methods.

5) Methods: Interrater reliability: Did more than one person analyze studies? Were intra and inter-rater reliability controls performed to ensure accurate analysis?

All studies were analyzed by the corresponding author. We have modified the Methods section to reflect this information.

6) Methods: "Organismal mammalian work" - This label is vague and could be divergently interpreted by different readers. Does this just mean whole organism behavioral analysis? Does it include all mammals, including primates and humans, or just rodents? Beyond answering these specific questions, the entire a priori article section criteria needs to be much better documented. This documentation should include how the employed protocol may potentially skew findings.

We updated the Methods section to include a detailed listing of the eligibility criteria for each article included in this study.

7) Methods: "Whether data were analyzed be sex." This definition is vague and requires further documentation. What analyses were considered an analysis by sex? For example, if sex was employed as a covariate, was this considered a sex analysis? What if the data or analysis/statistics were not shown? What if data were presented as disaggregated by sex but not further analyzed? Similar to the previous point, this section requires more robust documentation so that the reader can understand how this study was conducted.

We thank the authors for this comment and have updated the Methods section to reflect a more accurate description of sex-based analyses.